# Effect of Spinal Alignment Changes on Lower Back Pain in Patients Treated with Total Hip Arthroplasty for Hip Osteoarthritis

**DOI:** 10.3390/medicina57111219

**Published:** 2021-11-09

**Authors:** Fumiko Saiki, Takeyuki Tanaka, Naohiro Tachibana, Hirofumi Oshima, Taizo Kaneko, Chiaki Horii, Hideki Nakamoto, So Kato, Toru Doi, Yoshitaka Matsubayashi, Yuki Taniguchi, Sakae Tanaka, Yasushi Oshima

**Affiliations:** Department of Orthopaedic Surgery, The University of Tokyo, Japan 7-3-1 Hongo, Bunkyo-ku, Tokyo 113-8654, Japan; saikifum@yahoo.co.jp (F.S.); tatanaka-tky@umin.ac.jp (T.T.); hiro07183004@yahoo.co.jp (N.T.); hiro-o@ra3.so-net.ne.jp (H.O.); KANEKOT-ORT@h.u-tokyo.ac.jp (T.K.); harukabc@gmail.com (C.H.); nakamoto-tky@umin.ac.jp (H.N.); skatou-tky@umin.net (S.K.); tooru_doi@yahoo.co.jp (T.D.); matsubayashiy-ort@h.u-tokyo.ac.jp (Y.M.); taniguchiy-ort@h.u-tokyo.ac.jp (Y.T.); tanakas-ort@h.u-tokyo.ac.jp (S.T.)

**Keywords:** lower back pain (LBP), total hip arthroplasty (THA), alignment, patient-reported outcomes (PROs)

## Abstract

*Background and objectives*: The influence of changes in spinal alignment after total hip arthroplasty (THA) on improvement in lower back pain (LBP) remains controversial. To evaluate how changes in spinal malalignment correlate with improvement in preoperative LBP in patients who underwent THA for hip osteoarthritis. *Materials and Methods*: From November 2015 to January 2017, 104 consecutive patients who underwent unilateral THA were prospectively registered. Whole spine X-rays and patient-reported outcomes (PROs) were obtained preoperatively and 12 months postoperatively. The PROs used were the Numerical Rating Scale (NRS) for back pain, EuroQol 5 Dimension, and Short Form-12. *Results*: Seventy-four (71%) patients with complete data were eligible for the analysis. The sagittal parameters changed slightly but significantly. Coronal alignment significantly improved. Twenty-six (37%) patients had LBP preoperatively. These patients had smaller lumbar lordosis (LL), larger PT, and larger PI minus LL than the patients without LBP. Fourteen (54%) of the 26 patients with preoperative LBP showed pain improvement, but there were no significant differences in the radiographic parameters. *Conclusions*: Although preoperative LBP was likely to be resolved after THA, there were no significant correlations between alignment changes and LBP improvement. The cause of LBP in patients with hip osteoarthritis (OA) patients might be multifactorial.

## 1. Introduction

It is well established that the maintenance of an upright standing position requires certain correlations among the spine, pelvis, and lower extremities [1,2,3,4,5]. For example, when a person is standing, the pelvis tilts anteriorly and lumbar lordosis increases. Therefore, when the lumbar spine becomes stiff with aging, leading to a consequent decrease in lumbar lordosis, the acetabulum tilts anteriorly, and more flexion is required of the hip joint. However, in patients with osteoarthritis (OA) of the hip joint, contracture of the hip joint can cause anteversion of the acetabulum, shortening of the affected limb, obliquity of the pelvis, and spinal sagittal and coronal malalignment [6].

Total hip arthroplasty (THA) is an established procedure for patients with hip OA that effectively relieves pain and restores function. Because the contracture of the hip joint is corrected by the surgery, it would be reasonable to speculate that the spinal sagittal alignment will also change [7,8]. In addition, preoperative leg length discrepancy is expected to improve, which may affect pelvic obliquity and scoliosis [9,10]. However, the influence of spinal alignment changes on clinical symptoms, such as lower back pain (LBP), remains controversial [11,12]. In this study, we aimed to evaluate the associations among spinal alignment changes and improvement in preoperative LBP after unilateral THA.

## 2. Materials and Methods

From November 2015 to January 2017, a total of 104 consecutive patients, who underwent unilateral THA at our university hospital, were prospectively enrolled. All the participants provided written informed consent before participating in the study, which was approved by the Institutional Review Board at the authors’ institution (IRB approval number 10965-1, Tokyo, Japan). All patients had been diagnosed as having severe symptomatic unilateral hip joint OA and underwent THA during the enrollment period.

Of the 104 consecutive patients enrolled, 27 patients who had incomplete data on the questionnaire were excluded. Furthermore, 1 patient was lost to follow-up within 12 months after surgery, 1 patient had dislocation of the hip joint, and 1 patient withdrew informed consent. Therefore, 74 (71%) patients were eligible for the analysis.

THA surgery was performed by senior hip surgeons at our institution using a posterior approach. All patients received cementless THA and underwent the routine thromboprophylaxis regimen and postoperative rehabilitation program.

Each patient underwent assessment of their whole spine, via standing X-ray radiographs before surgery and 1 year after surgery. Radiographic sagittal parameters included measurements of pelvic incidence (PI), pelvic tilt (PT), sacral slope (SS), lumbar lordosis (LL), and the distance between the C7 plumb line and the posterior corner of the sacrum (C7-SVA). Radiographic coronal parameters included measurements of the distance between the C7 plumb line and the central sacral vertical line (C7PL-CSVL) and the pelvic obliquity angle, which was defined as the angle between the line connecting the bilateral iliac crests and a horizontal line (Figure 1). The first author performed all radiographic measurements.

Patients were asked to complete questionnaires before surgery and at 1-year intervals after surgery. The patient-reported outcomes (PROs) used were the Numerical Rating Scale (NRS) for back pain, EuroQol 5 Dimension (EQ-5D), and Short Form-12 (SF-12). An NRS of ≥4 was defined as the presence of LBP, and improvement supported a change of by ≥2 was defined as the improvement in LBP [13]. We compared radiographic parameters between patients with and without an improvement in LBP after THA.

SPSS v25 (SPSS Software, IBM Corp., Armonk, NY, USA) was used to perform the Wilcoxon signed-rank test and Mann–Whitney U test. A *p* value of <0.05 was regarded as indicative of statistical significance.

## 3. Results

The mean age of patients at surgery was 62 years (28–83 years), and 63 patients (85%) were women. Regarding the hip joint on the contralateral side, 28 patients had mild OA, nine patients had severe OA, and 24 patients had undergone THA for the contralateral side. No patient underwent bilateral THA at the same time. The pre- and postoperative radiographic parameters are summarized in Table 1. The sagittal spinal parameters changed slightly but significantly, PI decreased, PT increased, and SS decreased. The coronal parameters, specifically, C7-CSVL and the pelvic obliquity angle, significantly improved after surgery. In the PROs, the physical component summary (PCS) in the SF-12 and EQ-5D significantly improved postoperatively (Table 1).

Twenty-six (37%) patients had LBP before surgery, whereas 48 patients did not. Patients with preoperative LBP showed smaller LL, larger PT, and larger PI minus LL than the patients without preoperative LBP (Table 2). In the 26 patients with preoperative LBP, the degree of LBP significantly decreased after surgery, with NRS values ranging from 6.0 to 4.8 (*p* < 0.01). Of these patients, 14 (54%) showed improvement supported by ≥2 changes in the NRS; however, there were no significant differences in the pre- and postoperative radiographic parameters (preoperative: Table 3, and postoperative: Table 4).

## 4. Discussion

We sought to assess the changes in spinal alignment and LBP, as well as the connections between spinal alignment and LBP, after THA. In general, coronal alignment significantly improved, whereas sagittal parameters changed only slightly. About half of the patients with preoperative LBP showed improvement postoperatively; however, we did not find any relationship between improvement in LBP and either sagittal or coronal alignment changes.

Hip-spine syndrome was originally described by Offierski and MacNab more than 3 decades ago [1]. The original concept of this syndrome was based on the fact that patients with hip OA experienced pain relief in the back after being treated for hip OA. The researchers showed that the flexion contracture of the hip joint led to increased pelvic forward tilt, lumbar lordosis, and, as a result, LBP. Because the contracture and the range of motion (ROM) of the hip joint improve after THA, it is reasonable to speculate that spinal alignment will change after THA. In this study, we showed that spino-pelvic sagittal parameters changed slightly after THA. PI slightly increased, although it is thought to be an individually constant value [14,15]. This finding was reasonable, considering that the center of the hip joint would shift caudally after THA in patients who experienced central migration of the femoral head preoperatively. Nevertheless, PT slightly increased, whereas SS slightly decreased, which reflected a reduced anteversion of the pelvis caused by the decreased contracture of the hip joint. These changes were consistent with previous findings, although the difference in the angles was small and might not be clinically significant.

The presence of LBP has been reported in patients with hip OA. According to previous reports, 21.2% to 56.5% of patients treated with THA had LBP before surgery [12,16,17,18,19], which was almost the same as reported in our study (36.6%). We speculate that the relatively wide range of incidence rates reported in the literature is the result of differences in the definition of LBP used among the studies Nevertheless, the incidence of LBP in patients with hip OA is considered relatively high. Moreover, many patients have shown pain relief in the lower back after THA, which accounts for 54% to 100% in the literature [7,12,16,17,18].

It is reasonable to speculate that changes in spino-pelvic alignment might reduce tension in the back muscles and relieve LBP [20,21,22]. However, the precise mechanism remains elusive. To explain why LBP is relieved after THA, two reports focused on spino-pelvic alignment changes before and after surgery. Weng et al. investigated the effect of THA on sagittal spinal alignment in 69 patients treated with THA [12]. In their study, 39 (56.5%) patients complained of LBP before surgery, 17 of whom reported complete resolution, and 22 of whom reported significant relief. Although the researchers concluded that the improvement in abnormal sagittal spinal-pelvic-leg alignment helped improve preoperative LBP, they did not show any difference in the radiographic parameters between patients with and without preoperative LBP. Eyvazov et al. investigated the effects of THA on spinal sagittal alignment and static balance in 28 patients [11]. They showed that LBP and the Oswestry Disability Index (ODI) significantly improved after surgery, but they did not find any significant correlations between postoperative changes in spinal sagittal alignment or postural balance and the improvements in LBP and ODI scores. Considering the results from these two reports, preoperative LBP improved to some extent after THA; however, the involvement of spinal sagittal malalignment with improvement in LBP remained uncertain.

It is well known that sagittal imbalance can cause LBP [23,24,25]. In our study, patients with preoperative LBP tended to show decreased LL and, consequently, a PI minus LL mismatch, as compared to those without LBP. However, although 54% of the patients with preoperative LBP showed improvement after THA, none of the spinal sagittal parameters were significantly correlated. Therefore, we assume that, although preoperative spinal sagittal malalignment might in part have affected the presence of preoperative LBP, other factors that change in the spinal sagittal alignment must have influenced this improvement in LBP. Our results do not necessarily eliminate the possibility of an effect caused by slight changes in the sagittal alignment because the number of patients with preoperative LBP was relatively small. Tiny changes in pelvic anteversion could have influenced the muscle tonus around the lumbar spine and pelvis.

Compared with the changes in spinal sagittal alignment, coronal balance improved noticeably after THA. This was expected because pelvic obliquity can be mostly corrected after THA as a result of improvement in the leg length discrepancy. It is well known that coronal imbalance can also cause LBP. Eguchi et al. reported that a reduction in scoliosis was correlated with an improvement in the Roland-Morris Disability Questionnaire (RDQ) scores in 30 patients undergoing THA [9]. Although we anticipated that the degree of improvement in coronal balance would affect LBP relief after THA, this effect was not observed in our study. We speculate that this might be because RDQ can be affected by disorders in the hip joint and in the lumbar spine.

Another possible explanation for LBP relief after THA is a change in the susceptibility to pain [26]. Patients with hip OA are always bothered by coxalgia, which could lead to hypersensitivity to pain. In this study, the patients whose preoperative LBP did not improve after THA showed worse quality-of-life outcomes in general, specifically on the EQ-5d and PCS. Although the postoperative mental component summary (MCS) of the SF-12 was not significantly different (*p* = 0.10), it is possible that physical and mental disorders related to the hip joint disorders might have affected the degree of LBP after THA in such patients.

This study has several limitations. First, the number of patients was relatively small, which made detailed statistical analysis impossible. Second, only one investigator measured the radiographic parameters in this study. Because measurement errors can occur in such cases with severe hip OA, examinations by two or three investigators would have increased the accuracy of the results. Third, we used a body figure printed on paper to show the specific locations of the pain; but, because of the close proximity of the hip and lower back, it may have been difficult for patients to completely distinguish between LBP and coxalgia. In such cases, the percentages of patients with preoperative LBP and of patients with improved LBP after surgery may appear to be higher than they are in reality. Fourth, we did not consider pain medications, which would have affected the pain status. Further investigation will be necessary to elucidate these problems. Fifth, we did not investigate ROM, and many of the medical records did not mention it, making it difficult to evaluate the relationship between ROM and pelvic tilt.

In conclusion, LBP was likely to be resolved after THA in our patients with hip OA. Although the spinal sagittal and coronal alignment certainly changed after surgery, we did not find significant correlations between alignment changes and LBP improvement. We speculate that changes in the pain threshold might have affected the degree of LBP; however, the underlying mechanism remained uncertain. The cause of LBP in patients with hip OA patients is considered to be multifactorial.

## Figures and Tables

**Figure 1 medicina-57-01219-f001:**
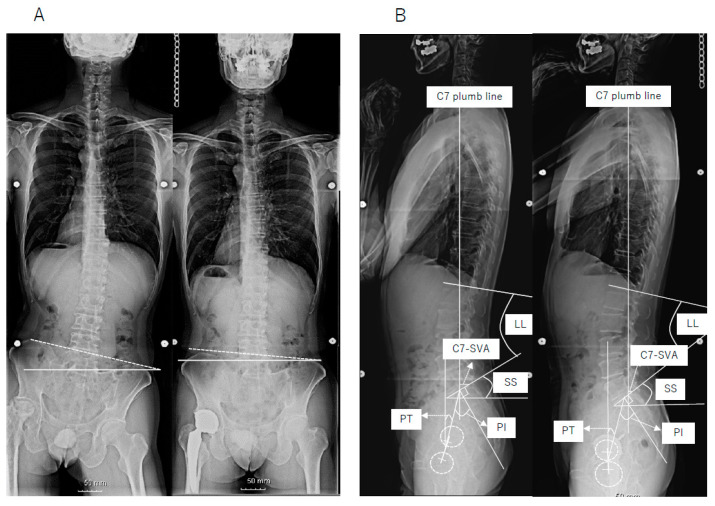
(**A**) Preoperative (left) and postoperative (right) frontal radiographs, showing pelvic obliquity angle, which was defined as the angle between the line connecting the bilateral iliac crests and a horizontal line. (**B**) Preoperative (left) and postoperative (right) lateral radiographs, showing spinopelvic parameters.

**Table 1 medicina-57-01219-t001:** Preoperative and postoperative radiographic parameters and patient-reported outcomes.

	Preoperative	Postperative (12 M)	*p*
	(*n* = 74)	(*n* = 74)	
**Radiographic Parameters**			
**Sagittal Parameters**			
C7-SVA	41.0 ± 43.1	37.1 ± 46.5	0.36
LL	51.9 ± 14.4	49.8 ± 16.6	0.08
PI	55.4 ± 10.1	53.5 ± 10.2	0.01
PT	15.6 ± 9.8	17.5 ± 9.9	0.01
SS	39.8 ± 8.3	36.1 ± 9.7	<0.01
PI minus LL	3.5 ± 15.3	3.7 ± 17.1	0.51
Coronal Parameters			
C7-CSVL	12.8 ± 10.6	7.6 ± 8.5	0.01
Pelvic Obliquity Angle	2.6 ± 3.0	1.6 ± 2.1	<0.01
Patient-Reported Outcomes			
LBP (NRS)	2.8 ± 2.3	2.4 ± 2.3	0.15
EQ-5D	0.74 ± 0.09	0.85 ± 0.10	<0.01
SF-12 PCS	28.5 ± 13.0	45.7 ± 12.9	<0.01
SF-12 MCS	54.2 ± 10.9	56.1 ± 8.6	0.13

Data are reported as mean ± SD. SVA indicates sagittal vertical axis; LL, lumbar lordosis; PI, pelvic incidence; PT, pelvic tilt; SS, sacral slope; CSVL, central sacral vertical line; LBP, lower back pain; NRS, numerical rating scale; EQ-5D, EuroQol 5 Dimension; SF-12, Short Form-12; PCS, physical component summary; MCS, mental component summary.

**Table 2 medicina-57-01219-t002:** Comparison of preoperative radiographic parameters and patient-reported outcomes between patients with and without preoperative lower back pain.

	LBP+ (*n* = 26)	LBP− (*n* = 48)	*p*
**Radiographic Parameters**			
**Sagittal Parameters**			
C7-SVA	54.8 ± 52.6	33.4 ± 34.9	0.26
LL	45.4 ± 18.2	55.5 ± 10.4	<0.01
PI	57.6 ± 10.3	54.2 ± 10.0	0.12
PT	19.9 ± 9.6	13.3 ± 9.3	<0.01
SS	37.7 ± 8.6	41.0 ± 8.1	0.13
PI minus LL	12.2 ± 18.8	−1.3 ± 12.4	<0.001
Coronal Parameters			
C7PL-CSVL	15.3 ± 11.9	11.4 ± 9.9	0.23
Pelvic Obliquity Angle	2.8 ± 2.8	2.5 ± 3.1	0.69
Patient-Reported Outcomes			
LBP (NRS)	6.0 ± 1.8	1.1 ± 0.9	<0.001
EQ-5D	0.73 ± 0.10	0.75 ± 0.08	0.31
SF-12 PCS	27.3 ± 12.0	29.2 ± 13.5	0.55
SF-12 MCS	52.4 ± 10.9	55.1 ± 10.9	0.49

Data are reported as mean ± SD. SVA indicates sagittal vertical axis; LL, lumbar lordosis; PI, pelvic incidence; PT, pelvic tilt; SS, sacral slope; CSVL, central sacral vertical line; LBP, lower back pain; NRS, numerical rating scale; EQ-5D, EuroQol 5 Dimension; SF-12, Short Form-12; PCS, physical component summary; MCS, mental component summary.

**Table 3 medicina-57-01219-t003:** Comparison of preoperative radiographic parameters and patient-reported outcomes in patients with preoperative lower back pain (improved vs not improved).

	Improved	Not Improved	*p*
	*n* = 14	*n* = 12	
**Radiographic Parameters**			
**Sagittal Parameters**			
C7-SVA	35.5 ± 54.0	64.0 ± 52.8	0.38
LL	44.5 ± 17.8	46.5 ± 19.4	0.98
PI	56.9 ± 11.7	57.5 ± 7.4	0.66
PT	20.7 ± 11.7	19.0 ± 6.8	0.54
SS	36.2 ± 7.3	38.5 ± 9.8	0.57
PI minus LL	12.4 ± 20.5	11.8 ± 17.2	0.98
Coronal Parameters			
C7PL-CSVL	15.4 ± 12.8	15.1 ± 11.2	0.88
Pelvic Obliquity Angle	3.1 ± 3.2	2.5 ± 2.2	0.62
Patient-Reported Outcomes			
LBP (NRS)	5.6 ± 1.9	6.3 ± 1.8	<0.001
EQ-5D	0.74 ± 0.11	0.72 ± 0.1	0.81
SF-12 PCS	29.2 ± 13.5	25.3 ± 10.2	0.42
SF-12 MCS	53.6 ± 12.8	51.0 ± 8.7	0.54

Data are reported as mean ± SD. SVA indicates sagittal vertical axis; LL, lumbar lordosis; PI, pelvic incidence; PT, pelvic tilt; SS, sacral slope; CSVL, central sacral vertical line; LBP, lower back pain; NRS, numerical rating scale; EQ-5D, EuroQol 5 Dimension; SF-12, Short Form-12; PCS, physical component summary; MCS, mental component summary.

**Table 4 medicina-57-01219-t004:** Comparison of postoperative radiographic parameters and patient-reported outcomes in patients with preoperative lower back pain (improved vs not improved).

	Improved	Not Improved	*p*
	*n* = 14	*n* = 12	
**Radiographic Parameters**			
**Sagittal Parameters**			
C7-SVA	58.3 ± 57.3	66.9 ± 49.3	0.57
LL	41.2 ± 21.2	41.7 ± 24.1	0.92
PI	56.5 ± 10.0	53.5 ± 10.1	0.90
PT	22.1 ± 10.4	21.5 ± 7.3	0.86
SS	34.4 ± 8.6	32.1 ± 10.7	0.50
PI minus LL	14.9 ± 22.4	13.0 ± 21.4	0.88
Coronal Parameters			
C7PL-CSVL	12.0 ± 9.8	10.6 ± 9.9	0.83
Pelvic Obliquity Angle	0.9 ± 0.9	2.4 ± 3.7	0.65
Patient-Reported Outcomes			
LBP (NRS)	2.4 ± 2.0	6.8 ± 2.3	<0.001
EQ-5D	0.86 ± 0.13	0.76 ± 0.10	0.04
SF-12 PCS	47.2 ± 13.3	36.7 ± 13.2	0.06
SF-12 MCS	56.6 ± 9.8	51.7 ± 7.6	0.10

Data are reported as mean ± SD. SVA indicates sagittal vertical axis; LL, lumbar lordosis; PI, pelvic incidence; PT, pelvic tilt; SS, sacral slope; CSVL, central sacral vertical line; LBP, lower back pain; NRS, numerical rating scale; EQ-5D, EuroQol 5 Dimension; SF-12, Short Form-12; PCS, physical component summary; MCS, mental component summary.

## Data Availability

The data presented in this study are available on request from the corresponding author. The data are not publicly available due to patient privacy.

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
