# Peer review of "Effect of Spinal Alignment Changes on Lower Back Pain in Patients Treated with Total Hip Arthroplasty for Hip Osteoarthritis"

_medicina, 2021, doi:10.3390/medicina57111219_

Round 1
Reviewer 1 Report
Please in the introduction and in materials and method, put the significance of LL, PI,PT, etc, not only in the table caption.
please make some minor revisions of English (es: line 178 "another factor that changes" or "other factors that change"; line 202 "in such cases with severe hip AO")
I suggest you amplify your article by citing these reports:
- Is it time for international guidelines on physical restraint in psychiatric patients? doi: 10.7417/CT.2019.2110
- Autopsy findings in COVID-19-related deaths: a literature review
Reviewer 2 Report
Thank you for a well-presented good study. The effect of a THA on spinal problems is still the focus of discussions.
Whilst as you correctly state the numbers in the study are small, there seems to be an indication that LBP improves. Would the return of ROM in the hip and hence the need for less compensatory ROM in the L-spine be of significance. This study, I would suggest, calls for a further study with more participants, maybe only patients with unilateral disease, inter-observer checks and sitting and standing X-rays pre-and postop.
I would like to see in the discussion/conclusion section how does study can provide a catalyst for more in-depth studies.
Reviewer 3 Report
Overall, an interesting article that sheds light on a context that is often forgotten or left unexamined. The work could be methodologically a bit better set up. I would recommend addressing the following issues:
- please use consistent abbreviations (e.g. THR in line 29, LPB in line 209)
- Line 36: please correct the wording "...have been is well..."
- role of posterior approach for THA vs. other approaches should be discussed in some way
- Please explain why patient acquisition was completed in 2017 and only now, 4 years later is publication being sought?
- The biggest problem for me is the insufficient differentiation of pain perception. I would have liked to see separate data on pain intensity in the affected hip area AND the lumbar area for all patients in this work, with a statement about which of these patients perceived back pain as significant. If these data are not available, I would recommend discussing this as a limiting factor. In the present version, this issue is addressed in lines 147 to 154, but it should be made clear here how difficult it can be to separate the perception of pain in the hip and lumbar regions in individual cases and how this may affect the conclusions of this study.
